# Tribological Behavior of Biomass Fast Pyrolysis Fuel and Diesel Blends

**Ruhong Song \*, Huiqiang Yu and Hui Song**

School of Mechanical Engineering, Hefei University of Technology, Hefei 230009, China; 1982800074@hfut.edu.cn (H.Y.); songhui@hfut.edu.cn (H.S.)
* Correspondence: 1984800043@hfut.edu.cn

**Abstract:** The original biomass fast pyrolysis fuel was modified by an emulsification method to obtain emulsified biomass fast pyrolysis fuel with different proportions (the content of biomass fast pyrolysis fuel in the emulsified biomass fuel was 5 wt.%, 10 wt.%, and 20 wt.%, respectively). Taking commercial 0$^{\#}$ diesel as the blank fuel, the friction and wear characteristics of tribo-pair material from an actual piston ring–cylinder liner lubricated by the varied fuels were investigated on a reciprocating friction and wear tester, respectively. The results showed that the friction coefficient and wear of tribo-pair material lubricated with emulsified biomass fuel increased with the biomass fuel content. In the case of 5 wt.% emulsified biomass fuel, the friction coefficient was smaller than 0$^{\#}$ diesel, and the wear was not different from 0$^{\#}$ diesel. At the same time, the friction coefficient and wear of 5 wt.% emulsified biomass fuel increased with the reciprocating frequency when the load was constant, while they increased with the load when the reciprocating frequency was constant.

**Keywords:** lubricity; biomass fuel; diesel; blend; friction and wear

## 1. Introduction

With the depletion of petrochemical energy, biomass fuel has a wide application prospect as a renewable new energy fuel, however, the quality of current biomass fuel cannot be compared with traditional fossil fuels. Therefore, it is necessary to improve the quality of original biomass fuel in order to be better used as fuel oil in traditional power output equipment such as internal combustion engines [1]. However, due to the high oxygen content, low cetane number, and high acid value of biomass fuel, it has low calorific value and strong corrosivity, and cannot be directly applied to internal combustion engines [2]. It can be applied only after physical and chemical modification, including blending with commercial diesel fuel.

There are many modification methods of biomass fuel in order to use biomass oil as an alternative fuel for engines. The emulsification method of biomass fuel/diesel is one of the promising modification technologies. Ikura et al. [3] prepared the microemulsion fuel with biomass fuel contents of 5–30%. The results showed that the corrosivity, calorific value, and cetane number of biomass fuel oil were improved after emulsfication. The cetane number of emulsified fuel with biomass fuel content of 5% could reach the vehicle demand. Bertoli [4], Chiaramonti [5], and Mujtaba [6] studied the emulsified fuel obtained by mixing biomass fuel with diesel for simple treatment directly. Bai et al. [7] conducted an engine bench test of the emulsified 10% and 15% biomass fuel/diesel blends on ZS1110 diesel engine bench. The results showed that the emulsified fuel with 15% biomass fuel volume fraction had an obvious fuel-saving effect than 0$^{\#}$ diesel fuel, and the maximum fuel saving rate could reach 10%. Both NO and CO emissions were also better than those of pure diesel. However, on the other hand, the engine fuel also requires a certain lubricity, mainly because the fuel needs to be transported by the oil pump during the fuel transportation in the engine. At the same time, the fuel will also dilute the lubricating oil of the internal

combustion engine. Therefore, the quality of the engine fuel lubricity has been a focus of many researchers.

Fazal et al. [8] aimed to assess the effect of propyl gallate additive on the sustainability and lubrication behavior of B30 (30% biodiesel in diesel) blend on mild steel flat surface by using a high-frequency reciprocating rig. The tribological results indicated that propyl-gallate-doped B30 provided better lubricating performance than other tested additives for steel/steel contacts. The presence of propyl gallate caused the least weight loss (0.0003 g) with the least wear scar width (1.13 mm). The average coefficient of friction was also observed to be minimal for the propyl-gallate-doped B30 blend. The compounds formed on the mild steel surface when tested with propyl-gallate-doped B30 showed relatively less oxygen and high carbon content. The possible mechanism in enhancing lubricity of propyl-gallate-doped B30 blend could be attributed to the formation of relatively more stable and effective ester-based tribo-films at the contact surfaces.

Ajayi et al. [9] studied the fuel dilution rate of three lubricating oils (E0, E10, and B16) in a marine engine operating in on-water conditions with a start-and-stop cycle protocol. The level of fuel dilution increased with the number of cycles for all three fuels. The most dilution was observed with B16 fuel, and the least with E10 fuel. In all cases, fuel dilution substantially reduced the oil viscosity. The impacts of fuel dilution and the consequent viscosity reduction in the lubricating capability of the engine oil in terms of friction, wear, and scuffing prevention were evaluated by four different test protocols. Although the fuel dilution of the engine oil had minimal effect on friction because the test conditions were under the boundary lubrication regime, significant effects were observed on wear in many cases.

Awang et al. [10] aimed to examine the diesel engine's performance and emission of secondary fuels (SFs) comprising waste plastic oil (WPO) and palm oil biodiesel (POB), and to analyze their tribological properties. Five SFs (10–50% POB in WPO) were prepared by mechanical stirring. The results showed that the secondary fuels had great influence on the diesel engine's performance.

In general, advanced alternative fuels provide an essential mechanism for reaching global aspirations of peak $CO_2$ emissions and net-zero carbon, driving research and development. There are many opportunities for the development of sustainable fuels, but equally many challenges [11,12]. It is important for the selection and application of biofuel in diesel, in which it is key data for the lubricity of biofuel and diesel blends. In the present study, the rice husk rapid pyrolysis fuel made by our research group was selected, and the biomass fuel was modified by a simple and practical emulsification method to improve the quality of biomass fuel. The lubrication characteristics of emulsified mixed oil (or emulsified biomass fuel) of biomass fuel and diesel with different ratios (5 wt.%, 10 wt.%, and 20 wt.%) were discussed, which was expected to provide basic data for the future application of the biomass fuel in engines.

## 2. Experimental

### 2.1. Material and Instrument

Commercial $0^{\#}$ diesel (SINOPEC, China Petrochemical Corporation, Beijing, China). Fast pyrolysis of rice husk biomass oil and SP emulsifier (HLB 5.0) was provided by our research group. Acetone (AR, Shanghai Zhenqi Chemical Reagent Co., Ltd., Shanghai, China). Multifunctional piston ring–cylinder liner simulation tribometer (detailed below). High shear laboratory emulsifying machine (SG 400, Shanghai Shanggui Fluid Equipment Co., Ltd., Shanghai, China). Scanning electronic microscope (SEM), and energy-dispersive X-ray spectrometry (EDS) (JSM-6490LV, Japan Electronics Corporation, Tokyo, Japan).

### 2.2. Structure and Principle of Tribometer

The experiment was carried out on a multifunctional piston ring–cylinder liner friction and wear simulation tester. Its structural diagram is shown in Figure 1. The device was composed of a driving part, loading part, friction measurement part, real-time temper-

ature measurement part, etc. The model of the temperature sensor was PT100/PT1000 (−200–650 °C). The driving part of the testing machine was realized by the crank slider mechanism. A long rod and fixture were fixed on the piston to drive the upper and lower piston ring specimens to slide relative to the upper and lower cylinder liner specimens. The crank length was 57 mm, the connecting rod length was 175 mm, and the stroke of the testing machine was 80 mm. The piston ring sample was cut from the actual piston ring by wire cutting, with a diameter of 110 mm, a circumferential width of 10 mm and a piston ring height of 3 mm. The cylinder liner (122 mm × 15.6 mm × 6.3 mm) sample was cut from the actual cylinder liner by wire cutting. The measurement of friction force was realized by a tension and pressure sensor, TJL-1 tension pressure sensor, the comprehensive precision was 0.02–0.05% F·S, and the wear loss was obtained by weighing with an electronic balance. A high-quality boron alloy cylinder liner was provided by Yangzhou Wuqiao Co., Yangzhou, China. The piston ring (double chromium ductile iron) was from Nanjing Feiyan Piston Ring Co., Nanjing, China.

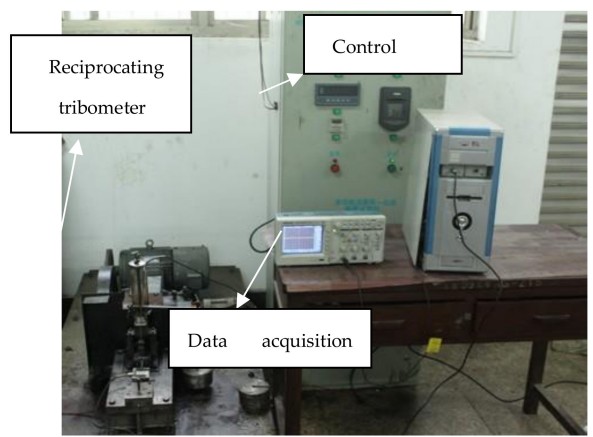 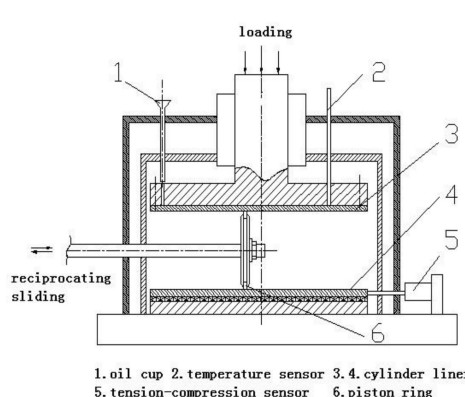

**Figure 1.** The sketch map of the piston ring–cylinder liner in the reciprocating engine.

According to the actual contact state between the piston ring and cylinder liner of diesel engine, the test load was selected to 210 N [13], test stroke 80 mm, test time 6 h, and the oil supply method was oil dropping lubrication (oil dropping speed: 25 mL/h). In order to ensure the reliability of test results, the friction and wear test was repeated three times according to the same specification under the same conditions. In this experiment, the friction and wear characteristics of microemulsion biomass fuel at different crank speeds (300 r/min, 450 r/min, and 600 r/min), i.e., reciprocating frequency of 5 Hz, 7.5 Hz, and 10 Hz, were investigated and compared with commercial 0[#] diesel under a load of 210 N and reciprocating frequency of 10 Hz.

### 2.3. Data Acquisition

The measurement of friction force was complex. Firstly, the pressure electrical signal was obtained by the tension and pressure sensor, and then the information was collected by the digital oscilloscope after amplification, filtering, and shaping by the control circuit, and the voltage information value was recorded according to the formula $F = 20 \times U$, where $U$ was the voltage, in V, and then converted to frictional force.

Friction coefficient was obtained according to formula $\mu = F/N$, where $F$ was the frictional force, in N. $N$ was the load in N. The wear loss $\Delta M$ ($\Delta M = M_1 - M_0$) was measured by the weight loss method. Before the test, the sample was ultrasonically cleaned with acetone for 10 min and dried at the oven temperature of 110 °C for 30 min, then weighed with an electronic balance with an accuracy of 0.1 mg and the initial mass, $M_0$, was recorded. The weighing process of the sample after the test was consistent with that before the test, and the mass was recorded as $M_1$. All experimental data come from the average of three measurements.

## 3. Results and Discussion

### 3.1. Preparation of Emulsion Biomass Fuel Blends

An amount of 5 wt.% emulsified biomass fuel, or biomass fuel/diesel emulsified blend, was prepared based on our previous method [14] by taking 5 wt.% of biomass oil, 89.5 wt.% of $0^{\#}$ diesel, and 5.5 wt.% of surfactant (composed of 93.1 wt.% of lipophilic surfactant and 6.9 wt.% of hydrophilic surfactant). The detailed preparation process was to add lipophilic surfactant to $0^{\#}$ diesel, respectively, hydrophilic surfactant to biomass fuel, and then use a high shear emulsifying machine to mix at 30 °C with 1500 r/min. By shearing and stirring the mixture of the above two components for 20 min, a stable emulsified biomass fuel was prepared, and its stability time could be up to over 60 days. The appearances of the fuels are shown in Figure 2. According to the above methods and process conditions, 10 wt.% and 20 wt.% biomass oil were taken, respectively, and appropriate additives were selected to prepare emulsified biomass fuel with different biomass fuel contents. Their optimum process conditions are shown in Table 1.

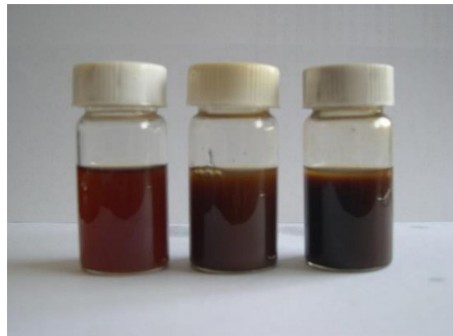

**Figure 2.** Appearance of different emulsified biomass fuels.

**Table 1.** Optimal preparation parameters of varied biomass fast pyrolysis fuel and diesel blends.

| Emulsion Biomass Fuel | Temperature (°C) | Stirring Time (min) | Additive Content (wt.%) | Rotation (r/min) |
|---|---|---|---|---|
| 5 wt.% | 30 | 20 | 5.5 | 1500 |
| 10 wt.% | 65 | 80 | 3.0 | 1500 |
| 20 wt.% | 65 | 120 | 4.0 | 1500 |

Du et al. [15] reported that the emulsified biomass fuel has good stability when the mass fraction of biomass fuel was not more than 20% and the stability time was more than 5 days. When the mass fraction was reduced to 10%, the stability time was more than 17 days, and when the mass fraction was reduced to 5%, the stability time would be more than 40 days. If the mass fraction of biomass fuel reached 30%, the stability time did not exceed 3 h, which might be due to one or several unstable factors in biomass fuel affecting the stability of emulsified biomass fuel. In addition, when the mass fraction of biomass fuel was greater than 20%, the viscosity of emulsified fuel would exceed the upper limit of vehicle diesel requirements ($8mm^2$/s, 20 °C). In summary, when the emulsified biomass fuel was applied to the engine, the mass fraction of added biomass fuel should not exceed 20%. Therefore, this study investigated the physicochemical and tribological performances of emulsified biomass fuel with biomass fuel content less than 20 wt.%. As shown in Figure 2, the appearance color of emulsified biomass fuel becomes darker changed from red-brown to dark brown with the increase in biomass fuel mass fraction.

### 3.2. Physicochemical Performances of Emulsion Biomass Fuel Blends

The basic physical and chemical properties of emulsified biomass fuel with different biomass fuel contents were shown in Table 2. The related measuring devices or references are detailed in reference [12].

**Table 2.** Basic performance of varied biomass fast pyrolysis fuel and diesel blends.

| Items | 0# Diesel | Biomass Fuel | 5 wt.% Emulsion Biomass Fuel | 10 wt.% Emulsion Biomass Fuel | 20 wt.% Emulsion Biomass Fuel |
|---|---|---|---|---|---|
| C (m/m)/% | 85.55 | 32.35 | 82.42 | 80.05 | 75.58 |
| H (m/m)/% | 13.49 | 8.36 | 13.23 | 12.72 | 10.22 |
| O (m/m)/% | 0.66 | 58.06 | 3.99 | 6.83 | 13.56 |
| Calorific value/(MJ·kg) | 47.25 | 17.15 | 45.91 | 41.61 | 36.44 |
| Density (20 °C)/(kg·m$^3$) | 737.80 | 862.60 | 778.16 | 789.49 | 805.31 |
| Viscosity(40 °C) /(mm$^2$·s) | 2.71 | 2.10 | 3.09 | 3.33 | 3.91 |
| Freezing point/°C | −15 | <−58 | −14 | −16 | −19 |
| Surface tension (20 °C)/(mN·m) | 31.1 | 37.6 | 33.7 | 34.0 | 35.1 |
| Acid value /(mg·KOH·g) | 0.12 | 33.10 | 0.82 | 1.21 | 1.88 |
| Water (vol)/% | less | 24.50 | 1.35 | 2.45 | 4.42 |

It was shown from Table 2 that compared with 0# diesel, the density of modified biomass fuel, i.e., emulsified biomass fuel oil, was relatively low, with low calorific value and high acid value, which was consistent with the low density, low calorific value, and high acid value of biomass fuel oil itself. The viscosity of the original biomass fuel oil was lower than 0# diesel, but the viscosity of the oil after emulsification was higher, even higher than 0# diesel. With the increase in biomass fuel content, the oxygen content, density, viscosity, surface tension, acid value, and water content of emulsified biomass fuel increased, while the content and calorific value of C and H elements decreased. The basic physical properties of emulsified biomass fuel were greatly improved compared with the original biomass fuel. Therefore, the quality of biomass fuel could be improved through emulsification technology.

### 3.3. Tribological Properties

#### 3.3.1. Friction and Wear

Figure 3 shows the variation of friction coefficient of emulsified biomass fuel and wear loss of the piston ring–cylinder liner with biomass fuel content under the working conditions of load 210 N and reciprocating frequency 5 Hz.

As shown in Figure 3a, the friction coefficient of emulsified biomass fuel gradually increased with the increase in biomass fuel content. When the biomass fuel content reached 20 wt.%, the friction coefficient of emulsified biomass fuel was even greater than that of 0# diesel. This was because with the increase in biomass fuel content and when the content exceeded 10 wt.%, the stability of emulsified biomass fuel decreased with the increase in temperature in the friction process, and the thickness of adsorption film formed in the friction process was small, so the friction coefficient was also high. When the content of biomass fuel was 5 wt.%, the friction coefficient of emulsified biomass fuel was smaller than 0# diesel, that is, the antifriction performance was better. This is because emulsified biomass fuel belongs to a heterogeneous thermodynamic stable system [16], in which small biomass fuel droplets are easy to be destroyed in the friction process, and the polar groups in biomass fuel oil are easy to be adsorbed on the metal surface [17], which promotes its deposition on the friction pair surface and plays the role of antifriction and lubrication. Diesel is mainly non-polar hydrocarbon molecules containing carbon chains, which have weak adsorption capacity on the friction pair metal surface and are easy to be squeezed out of the friction surface in the friction process.

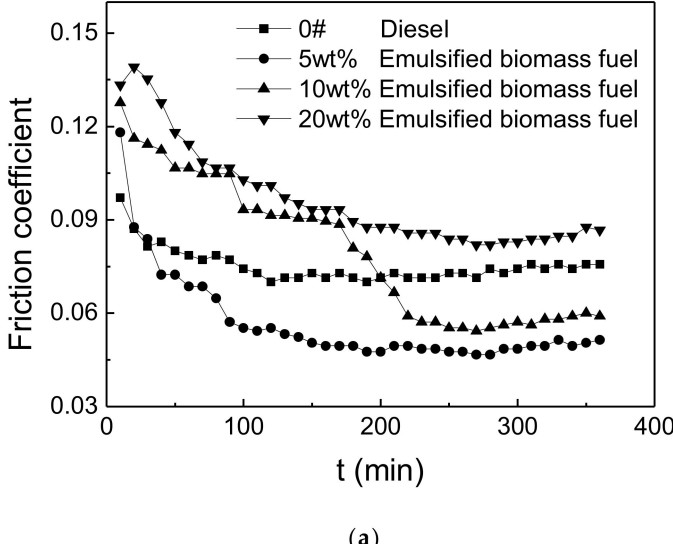

(**a**)

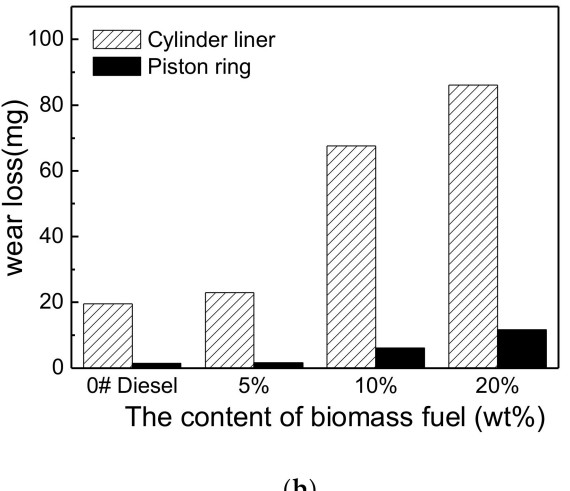

(**b**)

**Figure 3.** Changes of friction coefficient and wear loss of emulsified biomass fuel with biomass fuel content (**a**) friction coefficient and (**b**) wear loss.

As shown in Figure 3b, with the increase in biomass fuel content, the wear loss of cylinder liner and piston ring under emulsified biomass fuel dripping lubrication also increased. When the biomass fuel content exceeded 10 wt.%, the wear of the cylinder liner and piston ring increased sharply. This is because, on the one hand, complex chemical reactions occur in the friction heating process of biomass fuel, in which some substances are oxidized, resulting in the increase in acidic substances [18]. These acidic substances aggravate the corrosion and wear of friction pairs during the reciprocating movement of piston rings. On the other hand, with the increase in biomass fuel content, the oxygen content of emulsified biomass fuel also increases, and the existence of these oxygen elements may aggravate the corrosion and wear of the friction pair surface [19]. It can also be seen that the wear loss of the cylinder liner and piston ring under 5 wt.% emulsified biomass fuel dripping lubrication is slightly larger than that under 0# diesel dripping lubrication, in which the wear loss of the piston ring under 5 wt.% emulsified fuel lubrication is 1.6 mg and that of the cylinder liner is 23 mg. Under the lubrication of 0# diesel, the wear of the piston ring is 1.4 mg and that of the cylinder liner is 19.5 mg. This is because the oxygen element and some acidic substances in emulsified biomass fuel cause certain corrosion on

the surface of the piston ring cylinder liner during the reciprocating motion of the friction pair [20], and the corrosion will aggravate the wear and increase the wear loss.

In order to further explore the effect of emulsified biomass fuel with different biomass fuel content on the corrosion weight loss of cylinder liner materials, it was detected that the temperature of cylinder liner specimens lubricated by emulsified biomass fuel with different ratios was between 60 °C and 70 °C for a long time. Therefore, soaking experiments were carried out for emulsified biomass fuel with different ratios at 65 °C. Relevant experimental data are shown in Figure 4. It can be seen that with the increase in biomass fuel content, the oxygen content of emulsified biomass fuel increases and the acid value increases. When the immersion test was carried out for 120 h, the cumulative weight loss of the cylinder liner sample also increased, and the increase in oxygen content accelerated the corrosion effect of acid substances on friction pairs. The cumulative weight loss of the cylinder liner sample soaked in 5 wt.% emulsified biomass fuel for 120 h was slightly larger than that soaked in 0# diesel fuel.

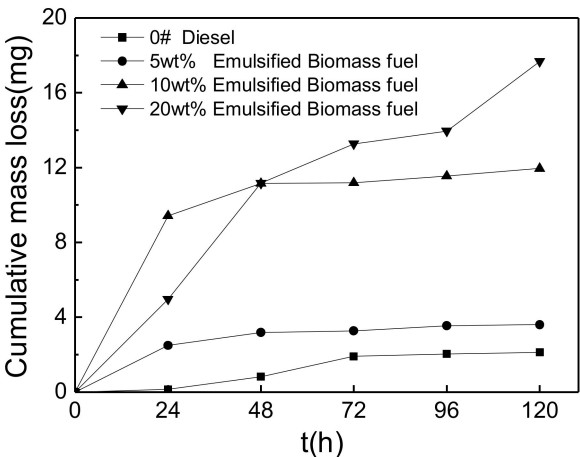

**Figure 4.** Soaking results of emulsified biomass fuel oil with different proportions.

From the above experiments, the following conclusions can be drawn: the antifriction performance of 5 wt.% emulsified biomass fuel is better than 0# diesel, and the wear resistance is equivalent to 0# diesel, that is, the lubrication performance of 5 wt.% emulsified biomass fuel is better than 0# diesel. Therefore, 5 wt.% emulsified biomass fuel is optimized to be selected as an alternative fuel to replace 0# diesel fuel in diesel engines.

### 3.3.2. Wear Mechanism

Figure 5 shows SEM images of the cylinder liner surface lubricated by emulsified biomass fuel with different ratios under the working condition of load 210 N and reciprocating frequency 5 Hz. It can be seen that the longitudinal oblique scratches are caused by a machining–honing process. With the increase in biomass fuel content in emulsified biomass fuel, more and more furrows (transverse) and corrosion pits appear on the cylinder liner surface, which may be due to the corrosion of metal caused by acidic substances produced by emulsified biomass fuel during friction. As a result, there are many irregular corrosion pits, and some corrosion products are scattered into the fuel during the reciprocating movement, which promotes the formation of more corrosion pits. With the increase in biomass fuel content in emulsified biomass fuel, the acid substances and oxygen content in the fuel increase, accelerating the corrosion of the base material, reducing the surface strength of the base material and loosening the texture. It is easy for the material to fall off and generate wear debris in the reciprocating process, and some wear debris accelerate the formation of furrows in the cylinder liner surface during the reciprocating process of the piston ring.

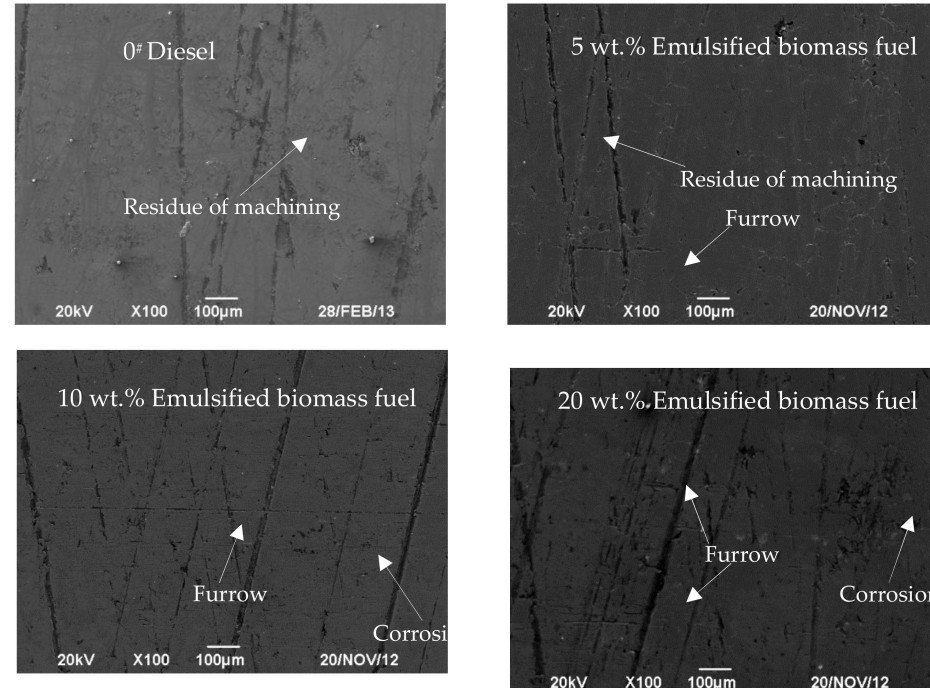

**Figure 5.** SEM images of the surface of the lower cylinder liner lubricated by the emulsified biomass fuel with different ratios.

As is well known, the chemical compositions of biofuel influenced the tribological behavior of diesel fuel. However, different preparation processes of biomass will cause varied chemical compositions of biofuel. The thermal decomposition of wheat straw at 350 °C and 450 °C was studied using the Py-GC/MS technique in a helium atmosphere to determine the gaseous compounds formed during biomass gasification under anaerobic conditions. The results obtained indicated the formation of compounds classified as phenols (vanillin, hydroxymethylfurfural) [21], which were similar with our previous results about biomass fast pyrolysis fuel [22]. In order to further explore the corrosion and wear characteristics of emulsified biomass fuel with different ratios under the above working conditions, EDS analysis related with the chemical composition of fuel was carried out on the wear mark surface of the cylinder liner, and the analysis results are shown in Table 3.

**Table 3.** EDS analysis results of the lower cylinder liner surface lubricated by emulsified biomass fuel with different proportions.

| Varied Blends | Element (wt.%) | | | | | | | |
|---|---|---|---|---|---|---|---|---|
| | C | O | Fe | Mn | Cr | Si | P | S |
| 5 wt.% | 1.86 | 6.39 | 86.07 | 1.06 | 0.36 | 3.43 | 0.81 | |
| 10 wt.% | 2.99 | 16.08 | 70.98 | 1.26 | 0.50 | 6.77 | 1.43 | |
| 20 wt.% | 7.25 | 24.78 | 55.26 | 1.07 | 0.43 | 9.98 | 0.67 | 0.56 |

Experimental conditions: load—210 N; frequency—5 Hz; time—6 h; and oil supply—dripping lubrication.

Table 3 shows the EDS analysis results of the cylinder liner surface lubricated by emulsified biomass fuel with different ratios. It can be seen that with the increase in biomass fuel content, the content of element C on the cylinder liner surface also increases. This is because the long carbon chain components and some C-containing organics in emulsified biomass fuel increase with the increase in biomass fuel content. These substances are adsorbed during friction and deposited on the friction surface through tribo-chemical reaction, which leads to an increase in the content of element C on the cylinder liner surface.

The content of the O element on the cylinder liner surface increases with the increase in biomass fuel content, but the content of the Fe element decreases. This may be because the existence of the oxygen element promotes the generation of corrosion wear. On the one hand, it accelerates the diffusion rate of the oxygen element to the inner layer of the metal surface, so that the detected content of the O element increases. On the other hand, it also accelerates the formation of metal oxide films such as $Fe_2O_3$ on the friction surface [23]. The thickness of the oxide film thickens with the increase in oxygen content so that the strength of the oxide film decreases and the brittleness increases, which is scraped off by the reciprocating piston ring to form abrasive particles. The falling off of abrasive particles reduces the content of Fe detected on the cylinder liner surface, and the falling off of abrasive particles is scattered in the oil. In the process of reciprocating friction, it intensifies the wear and makes the furrow deeper. This is basically consistent with the results detected in SEM.

**4. Conclusions**

(1) The physical and chemical properties of modified biomass fuel are greatly improved compared with the original biomass fuel; acid value, oxygen, and water content are reduced in varying degrees. The 5 wt.% modified biomass fuel with $0^{\#}$ diesel fuel blend basically meets the standard of vehicle fuel (GB 19147-2016).

(2) The corrosion and wear characteristics of reciprocating friction pairs are also different due to the different content of biomass fuel in emulsified biomass fuel. With the increase in biomass fuel content, its friction reduction becomes better, but its wear resistance becomes worse. In the case of 5 wt.% emulsified biomass fuel, its friction reduction is slightly better than that of $0^{\#}$ diesel, and its wear resistance is equivalent to that of $0^{\#}$ diesel.

(3) Some polar groups in the emulsified biomass fuel such as carboxylic acid are adsorbed on the metal rubbing surfaces to form a film with low shear strength, which prevents direct contact between metal surfaces and reduces the friction coefficient. On the one hand, with the increase in emulsified biomass fuel content the viscosity of lubricating oil will be reduced. On the other hand, some of the emulsified biomass fuels will produce some acidic substances during the friction process, which will cause corrosion on the surface of the friction pair and further increase the wear.

**Author Contributions:** Conceptualization, R.S.; and H.Y.; methodology, H.S.; validation, R.S.; formal analysis, H.Y.; investigation, H.Y.; resources, R.S.; data curation, H.S.; writing—original draft preparation, R.S.; writing—review and editing, R.S.; visualization, H.S.; supervision, H.S.; project administration, R.S.; funding acquisition, R.S. All authors have read and agreed to the published version of the manuscript.

**Funding:** This research received no external funding.

**Institutional Review Board Statement:** Not applicable.

**Informed Consent Statement:** Not applicable.

**Acknowledgments:** The authors wish to express their thanks to Y. F. Xu for his helpful discussions.

**Conflicts of Interest:** The authors declare no conflict of interest.

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
