# Peer review of "Tribological Behavior of Biomass Fast Pyrolysis Fuel and Diesel Blends"

_applsci, doi:10.3390/app12052540_

Round 1
Reviewer 1 Report
The paper is very interesting and well written.
I only suggest a more extensive analysis of the state of the art.
For example I suggesti the following papers.
Panepinto, D., Viggiano, F., Genon, G., Analysis of the environmental impact of a biomass plant for the production of bioenergy Renewable and Sustainable Energy Reviews, 2015, 51, pp. 634–647, 4586
Panepinto, D., Viggiano, F., Genon, G., The potential of biomass supply for energetic utilization in a small Italian region: Basilicata, Clean Technologies and Environmental Policy, 2014, 16(5), pp. 833–845
Author Response
Answer: I have added the papers mentioned above as references 11 and 12.
Reviewer 2 Report
The manuscript is about Tribological Behavior of Biomass Fast Pyrolysis Fuel and Diesel Blends. As for the characteristics of biomass, the literature review was limited, the authors should take into account similar studies that have been carried out in the discussion. The authors described the subject of biomass in a limited way. I recommend that you refer to biomass in more chemical aspects, I recommend reading the article: https://doi.org/10.3390/pr9020364 .For example in the above article, the PY-GC-MS of straw was shown.
It would be also good to describe the economic impact of the technology used and compare how much the commonly used technology (an example estimate by the used fertilizers) would cost to that proposed by the authors' technology. This is important because the cost of implementing the technology is the basis for its application.
Author Response
Answer: Thanks for your esteemed suggestions. I have cited the paper mentioned as reference 21, and discussed the fuel composition influence in the text (first paragraph on page 9).
Reviewer 3 Report
- Please provide more information about the sensors used in Fig. 1.
- It is recommended to use SI units for rotation, also in the tables. You have 1500 r/min in the text, but 1500 r/m in the Table 1.
- How many repetitions have been performed in the measurements?
- How were the physico-chemical properties of the fuels obtained in Table 2? There is no information on measuring devices or references.
- There are few errors in the text, where a “;” is used instead of a “.” at the end of a sentence. Please read the text again. For example: Line 13 (“piston rings; On the”), 19, 28 at page 7; Line 14, 20 at page 9, etc.
- The title of Fig. 4 has moved to another page. Please adjust.
- In conclusion 1, please indicate the standard to which the specific fuel conforms.
- Think about the measured values in context with the accuracy of the used measuring devices. Consider this when formulating conclusions.
Author Response
Detailed please see attachment. At the same time, as for the comment:
- The title of Fig. 4 has moved to another page. Please let editor do it.

Round 2
Reviewer 2 Report
The manuscript was prepered properly.